# Underwater Source Counting with Local-Confidence-Level-Enhanced Density Clustering

**DOI:** 10.3390/s23208491

**Published:** 2023-10-16

**Authors:** Yang Chen, Yuanzhi Xue, Rui Wang, Guangyuan Zhang

**Affiliations:** School of Microelectronics and Control Engineering, Changzhou University, Changzhou 213159, China; chenyang.heu@gmail.com (Y.C.); 15895064102@163.com (Y.X.); 20080902029@smail.cczu.edu.cn (G.Z.)

**Keywords:** acoustic vector sensor, source counting, DOA estimation, density clustering, local confidence level

## Abstract

Source counting is the key procedure of autonomous detection for underwater unmanned platforms. A source counting method with local-confidence-level-enhanced density clustering using a single acoustic vector sensor (AVS) is proposed in this paper. The short-time Fourier transforms (STFT) of the sound pressure and vibration velocity measured by the AVS are first calculated, and a data set is established with the direction of arrivals (DOAs) estimated from all of the time–frequency points. Then, the density clustering algorithm is used to classify the DOAs in the data set, with which the number of the clusters and the cluster centers are obtained as the source number and the DOA estimations, respectively. In particular, the local confidence level is adopted to weigh the density of each DOA data point to highlight samples with the dominant sources and downplay those without, so that the differences in densities for the cluster centers and sidelobes are increased. Therefore, the performance of the density clustering algorithm is improved, leading to an improved source counting accuracy. Experimental results reveal that the enhanced source counting method achieves a better source counting performance than that of basic density clustering.

## 1. Introduction

Acoustic vector sensors (AVSs) can co-pointedly and synchronously measure sound pressure and particle velocity vectors [1,2,3,4]. AVSs are widely used in underwater target detection, direction finding, tracking, and communication [5,6,7] due to their abilities of spatial filtering [8], DOA estimating [9], and resisting the interference of isotropic noise [10].

The target directing methods based on AVSs include the average sound intensity detector, sound intensity flow DOA histogram, cross-spectral DOA histogram, and so on [11,12]. AVSs can also be used in multi-channel arrays [13]. Each of these methods has its advantages and disadvantages. The histogram algorithm is widely used in engineering due to its better robustness compared with other algorithms. It can suppress narrowband and strong line spectrum interference and has a certain degree of multi-target resolution ability [14,15]. In [11], a DOA histogram based on sound intensity flow was used to estimate the DOA of targets, and the weighted DOA histogram of the line spectrum was used to realize the resolution and DOA estimation of multiple line spectrum targets. In [16], a DOA histogram was used to realize the resolution and DOA estimation of multiple wideband targets in the time–frequency domain and instantaneous frequency domain of the Huang transform. In an AVS sea trial experiment based on the Argo buoy platform in [17], it was also found that the DOA histogram could distinguish two wideband targets, the test ship and the engineering ship, whose adjacent interval was about 80 degrees. In reference [18,19], the windowed-disjoint orthogonality (WDO) of the target signal was further introduced to explain the mechanism of the wideband multi-target resolution ability of the DOA histogram.

The components of ship-radiated noise are very complex, including line spectrum, stationary ergodic random signals, and transient signals [20]. Therefore, it is considered to be broadband noise with energy distribution at any time and any frequency from the macro point of view. However, from the micro point of view, the energy intensity at different time–frequency points varies, which leads to a significant difference in the energy of different sources at some time–frequency points when multiple sources are synthesized at the sensor receiver, and a target playing a dominant role. This phenomenon is called the window-disjoint orthogonality (WDO) of signals and is widely used in blind signal separation for multi-source resolution and separation [21,22]. The higher the WDO characteristic of the signal in a certain time frequency point, the greater the energy of the dominant signal is compared to the sum of other signals, and the direction estimation result of the frequency point will be biased to the target direction of the dominant signal. If significant numbers of TF points possess the WDO property, in the DOA histogram the DOA estimates will cluster around the actual DOAs of the sources to achieve multi-target resolution of the histogram.

With multi-target resolution, it is possible to achieve source counting through multiple source detection. Traditional multiple source detection methods such as Minimum Description Length (MDL) [23], Akaike Information Criterion (AIC) [24], and Random Matrix Theory (RMT) [25] are overdetermined. They work well when the sensor channels outnumber the sources. However, for two-dimensional AVSs, these methods will fail when there are more than two sources, in which case the applicability of the AVS will be greatly limited.

As a result, an underdetermined method based on the density clustering algorithm was formed to solve the source counting problem for single AVSs in [18]. The density clustering algorithm was used to classify the DOAs of all the time–frequency points, where the cluster centers and the number of the clusters were obtained as the DOA estimations and target number, respectively. However, WDO is the inherent characteristic of the signal. With the increase in targets, the WDO decreases, which limits the multi-target resolution performance of the DOA histogram, and the performance of the density clustering algorithm deteriorated.

For this reason, a multi-source detection based on multimodal fusion was developed in [10]. The output of the AVS was decomposed into multiple modes by intrinsic time-scale decomposition (ITD), so that the source number in each mode decreased. For each mode, the WDO increased, leading to a better source counting accuracy. Therefore, the fused source counting performance was improved. However, the counting performance varied with the number of modes employed and it was difficult to determine the optimal number of employed modes.

In this paper, the local confidence level is adopted to weigh the density of each DOA data point to highlight the samples with the dominant source and downplay those without, so that the differences in densities for the cluster centers and sidelobes are increased. Therefore, the performance of the density clustering algorithm is enhanced. An analysis of lake trial data is conducted by comparing the proposed local-confidence-level-enhanced density clustering method with the methods in [10] and [18]. The results confirm the availability of the proposed local-confidence-level-enhanced density clustering method in improving source counting performance. The source counting accuracy of the proposed method is better than the methods it is compared with. The probability distribution of the source counting result for the local-confidence-level-enhanced method is more concentrated to the source number. As the SNR decreases, the proposed method undergoes slower degradation.

## 2. Model and Cross-Spectral DOA Histogram of Single-Vector Sensor

The output of an AVS contains information on both sound pressure and vibration velocity. The model of two-dimensional AVSs in the horizontal free sound field can be expressed as:(1)$$y(t)=\left[\begin{array}{c} P(t)\\ V_{x}(t)\\ V_{y}(t) \end{array}\right]=\left[\begin{array}{c} \sum_{n=1}^{N}x_{n}(t)+n_{p}(t)\\ \sum_{n=1}^{N}x_{n}(t)\cos\alpha_{n}+n_{x}(t)\\ \sum_{n=1}^{N}x_{n}(t)\sin\alpha_{n}+n_{y}(t) \end{array}\right]=\sum_{n=1}^{N}x_{n}(t)\left[\begin{array}{c} 1\\ \cos\alpha_{n}\\ \sin\alpha_{n} \end{array}\right]+e(t),$$
where N is the number of sources, P(t) is the sound pressure component of the AVS output, and $V_{x}(t)$ and $V_{y}(t)$ are the components of vibration velocity on two axes, respectively. $x_{n}(t)$ represents the signal radiated from the nth source to the AVS, $\alpha_{n}(t)$ indicates the angle of the nth sound source relative to the vector hydrophone (x axis is zero-degree orientation), $e(t)=[n_{p}(t),n_{x}(t),n_{y}(t{)]}^{T}$ is the noise vector, and $n_{p}(t)$, $n_{x}(t)$, and $n_{y}(t)$ are noises of the sound pressure, x-axis, and y-axis vibration velocities, respectively.

By taking the short-time Fourier transform (STFT) of each channel of the mixture, the AVS model can be expressed in the time–frequency domain as:(2)$$Y(\omega ,m)={\left[P(\omega ,m),V_{x}(\omega ,m),V_{y}(\omega ,m)\right]}^{T}=\sum_{n=1}^{N}X_{n}(\omega ,m)\left[\begin{array}{c} 1\\ \cos\alpha_{n}\\ \sin\alpha_{n} \end{array}\right]+E(\omega ,m),$$
where ω and m are the frequency bin and time frame indices, respectively. P(ω,m), $V_{x}(\omega ,m)$, $V_{y}(\omega ,m)$, $X_{n}(\omega ,m)$, and E(ω,m) are the STFTs of P(t), $V_{x}(t)$, $V_{y}(t)$, and e(t), respectively. The DOA estimation at each time–frequency point can be obtained by:(3)θ(ω,m)=∠[Re{P(ω,m)Vx*(ω,m)}+j⋅Re{P(ω,m)Vy*(ω,m)}],
where ∠ represents the phase angle, and $V_{x}^{*}(\omega ,m)$ and $V_{y}^{*}(\omega ,m)$ represent the complex conjugates of $V_{x}(\omega ,m)$ and $V_{y}(\omega ,m)$, respectively. The DOA histogram can be obtained by counting the estimation results of all time–frequency points. Supposing that the azimuth interval of histogram statistics is Δθ, we count the number of DOA estimation results in the interval [θ−Δθ/2,θ+Δθ/2]: when θ−Δθ/2<θ(ω,m)≤θ+Δθ/2, the amplitude of the histogram at the corresponding position is increased by 1:(4)$$H_{b}(\theta )=\sum_{\theta -\Delta \theta /2<\theta (\omega ,m)\leq \theta +\Delta \theta /2}1.$$

## 3. Local-Confidence-Level-Enhanced Density Clustering Source Counting

### 3.1. Density-Clustering-Based Source Counting

The density clustering algorithm depends on two important hypotheses: (1) The density for the center of a cluster is higher than that of the neighboring points; (2) The minimum distance between the center of a cluster and a higher-density point is relatively large [26,27]. The data set $\Theta =[\theta_{1},\theta_{2},\ldots ,\theta_{K}]$ is established with the DOA estimations obtained by Equation (3) in the time–frequency domain, where K is the number of the DOA estimations. Since the distance between two DOA estimations does not exceed $180^{\circ }$, the distance $d_{kl}$ between each two samples in Θ is defined as: (5)$$d_{kl}=\left\{\begin{array}{c} \left|\theta_{l}-\theta_{k}\right|,\;\;\;\left|\theta_{l}-\theta_{k}\right|<180\\ 360-|\theta_{l}-\theta_{k}|,\;\;\;\left|\theta_{l}-\theta_{k}\right|\geq 180 \end{array}\right.,\;k,l\in \left\{1,\cdots ,K\right\}$$
where $\theta_{k}$ and $\theta_{l}$ are two different samples in the data set Θ. They are angles limited within the interval of [−180°,180°]. The absolute difference between two angle values $\left|\theta_{l}-\theta_{k}\right|$ is within the interval of [0°,360°]. The distance between two angles is periodic with the period of 180°. So, when $\left|\theta_{l}-\theta_{k}\right|<180$, the distance $d_{kl}$ is $\left|\theta_{l}-\theta_{k}\right|$ itself. When $\left|\theta_{l}-\theta_{k}\right|\geq 180$, the distance $d_{kl}$ is $360-|\theta_{l}-\theta_{k}|$.

Then, the local density of each sample $\rho_{k}$ is calculated as:(6)$$\rho_{k}=\sum_{l\in \left\{1,\cdots ,K\right\}}\chi \left(d_{kl}-d_{c}\right),$$
where $\chi \left(x\right)=1$ if x<0 and $\chi \left(x\right)=0$ otherwise, and $d_{c}>0$ represents the cut-off distance used to keep a region for each sample. $\rho_{k}$ is the number of the samples within a limited distance of $d_{c}$ from the sample $\theta_{k}$, i.e., $\rho_{k}$ is equal to the number of points that are closer than $d_{c}$ to the sample $\theta_{k}$. The more samples there are near the sample $\theta_{k}$, the larger the local density of the sample $\theta_{k}$ is. $d_{c}$ can be chosen so that the average number of neighbors is around 1 to 2% of the total number of points in the data set [28]. A sample is assigned to the same cluster of its nearest sample with a higher density only if their distance is smaller than $d_{c}$. The minimum distance $\delta_{k}$ between the sample $\theta_{k}$ and any other sample with a higher density is measured as:(7)$$\delta_{k}=\underset{l:\rho_{l}>\rho_{k}}{\min}\left(d_{kl}\right).$$

For the point with the highest density, we conventionally take $\delta_{k}$ as:(8)$$\delta_{k}=\underset{l\in \left\{1,\cdots ,K\right\}}{\max}\left(d_{kl}\right).$$

The product $\gamma_{k}=\rho_{k}\delta_{k},k=1,2,\ldots ,K$ is used as the feature, and its value for a cluster center sample is obviously larger than that for the other sample. Thus, the differences among the ordered feature sequence are used to find the samples with significantly larger features, whose number corresponds to the number of clusters or targets [29]. In this procedure, the features $\gamma_{k},k=1,2,\ldots ,K$ are sorted in descending order, i.e.,
(9)$$\gamma_{1}\geq \gamma_{2}\geq \ldots \geq \gamma_{K}.$$

Suppose that there are L targets in the detection range of the AVS. Since the features of the cluster center samples are much larger than the other samples, the first L features $\left\{\gamma_{1},\gamma_{2},\ldots ,\gamma_{L}\right\}$ are significantly larger than the other K−L features $\left\{\gamma_{L+1},\gamma_{L+2},\ldots ,\gamma_{K}\right\}$. Consequently, the difference between $\gamma_{L}$ and $\gamma_{L+1}$ is relatively large. To obtain L, the differences in the ordered features are computed as:(10)$$\Delta \gamma_{i}=\gamma_{i}-\gamma_{i+1},\;i=1,2,\ldots ,K-1,$$

And the variance of the sequence ${\left\{\Delta \gamma_{i}\right\}}_{i=n}^{K-1}$ is calculated as:(11)$$\sigma_{n}^{2}=\frac{1}{K-n}\sum_{i=n}^{K-1}{\left(\Delta \gamma_{i}-\frac{1}{K-n}\sum_{i=n}^{K-1}\Delta \gamma_{i}\right)}^{2},$$
where n=1,2,…,K−2. Further, the second-order statistic of the features is defined as:(12)$$S_{n}=\{\begin{array}{cc} \frac{\sigma_{n+1}^{2}}{\sigma_{n}^{2}}, & \sigma_{n}^{2}>0\\ +\infty , & \sigma_{n}^{2}=0 \end{array}$$

Finally, the target (or cluster) number is estimated as:(13)$$L=\underset{n=1,\ldots ,K-3}{\arg\min}S_{n},$$

And the L DOA samples with the L largest features are obtained as the final multi-target DOA estimation results.

### 3.2. Local-Confidence-Level-Enhanced Density Clustering Algorithm

With the increase in sources, the WDO decreases, which limits the multi-target resolution performance of the DOA histogram. DOA estimations of different time–frequency points have different contributions to density clustering. For a certain time–frequency point, the larger the proportion of the dominant signal energy is, the closer the DOA estimation is to the truth value, and the greater the contribution this sample provides to the clustering. A local-confidence-level-based density enhancement algorithm is proposed in which the contribution of the samples with a high proportion of the dominant signal energy are exaggerated, so that the accuracy of the clustering and the precision of the target number estimation are improved.

For a certain time–frequency point (ω,m) (read circle) in the STFT domain, define the rectangular area $\Omega_{\omega ,m}$ (Dotted Box) around it, as illustrated in Figure 1. The width and length of this area are $l_{m}$ time points and $l_{\omega }$ frequency points, respectively. Therefore, there are $l_{\omega }l_{m}$ time–frequency points in the rectangular area $\Omega_{\omega ,m}$. The local confidence Γ(ω,m) of this time–frequency point (ω,m) is estimated by performing a principal component analysis (PCA) [30] on the snapshot vector $Y\left(\Omega_{\omega ,m}\right)$ in the region $\Omega_{\omega ,m}$, which reflects the strength of the dominant signal at the time–frequency point (ω,m).

For each time–frequency point, a snapshot $Y\left(\Omega_{\omega ,m}\right)$ is formed with all snapshots $Y\left(\omega ,m\right)$ in the region $\Omega_{\omega ,m}$. It is a $3\times l_{\omega }l_{m}$ matrix with columns Y(ω,m), $\left(\omega ,m\right)\in \Omega_{\omega ,m}$. A positive semidefinite complex Hermitian matrix is then constructed with $Y\left(\Omega_{\omega ,m}\right)$ for the region $\Omega_{\omega ,m}$ as:(14)$$R\left(\omega ,m\right)=Y\left(\Omega_{\omega ,m}\right)Y^{H}\left(\Omega_{\omega ,m}\right).$$

For a two-dimensional AVS, the rank of the matrix $R\left(\omega ,m\right)$ is 3. Performing an eigenvalue decomposition on $R\left(\omega ,m\right)$, we obtain three real-valued positive eigenvalues in decreasing order $\lambda_{1}(\omega ,m)\geq \lambda_{2}(\omega ,m)\geq \lambda_{3}(\omega ,m)$ of the matrix $R\left(\omega ,m\right)$. The local confidence level of the time–frequency point (ω,m) is expressed as:(15)$$\Gamma (\omega ,m)=\frac{2\lambda_{1}(\omega ,m)}{\lambda_{2}(\omega ,m)+\lambda_{3}(\omega ,m)}.$$

If there is only one dominant signal whose power is much larger than the noise in the time–frequency point (ω,m), $\lambda_{1}$ is the eigenvalue of the dominant signal which is much larger than $\lambda_{2}$ and $\lambda_{3}$, which are the eigenvalues of the noise. Otherwise, if there are two similar signals whose power is much larger than the noise, $\lambda_{2}$ will be much larger than $\lambda_{3}$, and the local confidence will become much smaller than the one dominant signal case. Therefore, it can be used to represent the WDO.

It is obvious that the local confidence level is proportional to the ratio of the dominant signal.

Then, the local confidence is used to enhance the cross-spectral DOA histogram. In the original orientation histogram statistics, the density of each sample is 1. The local-confidence-weighted DOA histogram with a sample density of Γ(ω,m) is:(16)$$H_{e}(\theta )=\sum_{\theta -\Delta \theta /2<\theta (\omega ,m)\leq \theta +\Delta \theta /2}\Gamma (\Omega_{\omega ,m}).$$

Substituting Equation (4) into Equation (16) yields:(17)$$H_{e}(\theta )=H_{b}(\theta )\cdot \left[\sum_{\theta -\Delta \theta /2<\theta (\omega ,m)\leq \theta +\Delta \theta /2}\Gamma (\Omega_{\omega ,m})\right]=H_{b}(\theta )\bar{\Gamma }(\theta ),$$
where $\bar{\Gamma }(\theta )$ is the average local confidence level.

The enhanced local density $\rho_{k}^{\prime }$ of the kth sample is weighted with the corresponding local confidence level Γ(k) as:(18)$$\rho_{k}^{\prime }=\Gamma (k)\rho_{k}.$$

The minimum distance is calculated according to the weighted local density $\rho_{k}^{\prime }$, and the target number is estimated by the ordered feature sequence afterward. The remaining steps are the same as in Section 3.1.

With the enhanced local density, the density of the sample points that are closer to the cluster center will increase while the density of the sample points far away from the cluster center will decrease. Consequently, the performance of the density-clustering-based multi-target detection is improved. To summarize, the proposed source counting method based on the local confidence-enhanced density clustering is depicted in Algorithm 1.
**Algorithm 1:** Local-confidence-level-enhanced density clustering source counting**Input:** Output y(t) of AVS.**Output:** Target number L and the DOAs (θ^1,θ^2,…,θ^L) of the targets.1 Compute the STFT Y(ω,m) of y(t) with Equation (2);2 Define the rectangle area Ωω,m for each (ω,m) point;3 Compute positive semidefinite complex Hermitian matrix R(Ωω,m) with Equation (14);4 Perform eigenvalue decomposition on R(Ωω,m) to obtain the eigenvalues λ1(ω,m), λ2(ω,m), and λ3(ω,m);5 Compute the local confidence level Γ(ω,m) with Equation (15);6 Compute the DOA estimations θ(ω,m) at all the time–frequency points with Equation (3);7 Consist data set Θ with the DOA estimations;8 **for** k=1 **to** K9   Compute the enhanced local density ρk′ of each sample with Equation (18);10   **for** l=1 **to** K11    Compute the distance dkl between each two samples in Θ with Equation (5);12   **end for**13 **end for**14 **for** k=1 **to** K15   Compute the minimum distance δk with Equations (7) and (8);16   Compute features as γk=ρk′δk;17 **end for**18 Sort the features γk (k=1,2,…,K) in descending order;19 Compute Δγi of the ordered features with Equation (10);20 Compute the variance of {Δγi}i=nK−1 with Equation (11);21 Compute the second-order statistic Sn of the ordered features with Equation (12);22 Estimate the target number L with Equation (13);23 Search DOAs (θ^1,θ^2,…,θ^L) with the L largest features.

## 4. Lake Trial and Analysis

### 4.1. Experimental Settings

The experimental data were collected in FuXian Lake. The experimental scenario is depicted in Figure 2, in which the location of a two-dimensional co-vibration AVS is in the middle of the lake. The sensitivity of the velocity channel decreased with increasing frequency (as the slope of −6 dB / octave), and there was a 90-degree phase difference between the vibration velocity channel and the sound pressure channel. The AVS was rigidly fixed about 4 m underwater on the side of the survey ship. The output signals of the AVS were collected and stored by a multi-channel synchronous data acquisition system. In the experiment, four ships (yachts) acted as moving targets located 1–2 km around the AVS. The ships were distributed in four directions of about 40°, 140°, 210°, and 330° at the beginning. The sampling rate was 48 kHz, the length of the STFT window was 8192 points, the frequency band was 0.5–8 kHz, the integral length of the azimuth histogram was 1 s (six windows), the sliding step was 8192 points (one window), and the rectangular area was $l_{m}=l_{w}=3$. The signals of two vibration velocity channels were compensated in the frequency domain according to the slope and phase of sensitivity after STFT.

### 4.2. Resolution Performance

Figure 3 illustrates the temporal course of the cross-spectral DOA histogram computed with Equation (4) at each time bin. Four targets marked as <1>, <2>, <3>, and <4> are presented in the histogram. The red color stands for the peaks of the histogram, which indicate the locations of the targets. Continuous peaks in time compose the time-bearing course of a target, which is illustrated roughly with straight black, red, green, and blue dash–dot lines, respectively, for each target. As shown in Figure 3, the azimuths of targets 3 and 4 gradually become closer over time, so the spectral peaks of these two targets become hard to distinguish (red color connects in one piece around time (d)). Since the signal from the target ship is weak when the ship decelerates or stops, the spectral peaks of targets 1 and 2 are unapparent or even submerged by the background during 38–40 s in the DOA histogram, as with the time points 44–50 s of target 3 and 50–53 s of target 4 (red color absent). Thus, the target courses have obvious discontinuity.

Figure 4 depicts the temporal course of the local-confidence-level-enhanced cross-spectral DOA histogram computed with Equation (16) at each time bin. It can be seen that the course of target 3 is significantly enhanced (red color is more continuous in time and peaks are higher than those in Figure 3), and the course of other targets is also enhanced relative to the original ones in Figure 3. It indicates that by using the local confidence level enhancement, the DOA courses of these targets become apparent, and both the resolution of targets and the continuity of the courses are improved (red color areas are thinner in bearing axis direction and more clearly separated compared to Figure 3). 

Figure 5 shows the comparative results between the cross-spectral DOA histogram and the enhanced cross-spectral DOA histogram at different time points corresponding to (a), (b), (c), and (d) in Figure 3 and Figure 4, respectively. As shown in Figure 5, both the original and the enhanced histograms present spectral peaks at the directions of the targets, while the enhanced one possesses glaringly obvious peaks. This reveals that the enhanced histogram is able to estimate the azimuths of targets at a higher resolution. In particular, when the signal from the target ship is weak, such as target 3 in Figure 5c, the enhanced cross-spectrum can significantly upgrade its peak, which makes it possible to detect weak targets. Additionally, as depicted in Figure 5d, the distinguishability of adjacent targets (targets 3 and 4) is also promoted by the enhanced cross-spectrum.

Figure 6 explains the enhancement performance of the local confidence level by the DOA histogram at time (c). The average local confidences of the directions of the targets are larger than the others. As a result, the peaks of the targets become higher and sharper.

### 4.3. Source Counting Performance

To explain the advantage of the local-confidence-level-enhanced density clustering algorithm, Figure 7 illustrates the decision graphs of the basic and enhanced density clustering algorithms with the data in Figure 6. Due to the inconspicuousness of the peaks of targets 1 and 3 (i.e., points c and d in Figure 6a), the cluster centers (i.e., decision points c and d in Figure 7a) of the basic density clustering are close to the cluster halos, which are recognized as the background. Nevertheless, in the enhanced density clustering algorithm, the local density is enhanced with the corresponding local confidence level which is represented by the size of the points in Figure 7. The local confidence level is higher around the cluster center than that in the cluster halo. Therefore, the background becomes more clustered and the cluster centers are far away from the background, especially for targets 1 and 3 (i.e., points C and D in Figure 7b). Therefore, the enhanced local density leads to a clearer distinction between the cluster centers and the background in the decision graph, which makes it easier to achieve correct counting.

Figure 8 demonstrates the second-order statistic $S_{n}$ of the basic and enhanced density clustering algorithms with the data in Figure 6. Obviously, the basic density clustering obtains the wrong counting result while the enhanced density clustering obtains the correct one. These results are consistent with the analysis of Figure 7.

Figure 9 illustrates the courses of the multi-target DOA estimations obtained by the basic density clustering and the local-confidence-level-enhanced density clustering. Compared with the courses estimated by the basic density clustering, the outliers in the courses estimated by the enhanced density clustering significantly reduce by weighting the density with the local confidence level, and the continuity of the courses of the multi-target detection is remarkably improved, especially in Regions A and B. Moreover, the distribution of the DOA estimations of targets is more compact for enhanced density clustering. 

Figure 10 depicts the results of the source number estimations obtained by the basic density clustering and the local-confidence-level-enhanced density clustering. Obviously, in Regions A, B, and C, the enhanced density clustering achieves the correct estimation results while the basic density clustering does not. In Region D, although both methods obtain incorrect results, the outcomes of the enhanced density clustering are closer to the true value. Targets 3 and 4 stop for a while during the periods of 44–50 s and 50–53 s, respectively. During these periods, there are only three targets in the field. As a result, the source number estimation result in the period before 44 s was used to calculate the source counting accuracy.

A comparison was conducted between the proposed local-confidence-level-enhanced density clustering method, the multimodal-fusion-based method in [10], and the basic density-clustering-based method in [18]. As shown in Figure 11, the basic density-clustering-based method obtained 48.82% accuracy of target number estimation, the multimodal-fusion-based method obtained an accuracy of 51.15% with four modes employed, while the local-confidence-level-enhanced method achieved 63.39% accuracy. The probability distribution of the source counting result for the local-confidence-level-enhanced method is more concentrated to the source number.

The multimodal-fusion-based method significantly reduces the probability of underestimating the errors of one and two sources and worsens the probability of overestimating errors of five sources. The WDO increases and sources are more detectable in each mode, so the probability of missed detections is reduced. However, a source signal may be divided into parts and be distributed in more than one mode. These parts may not be fully recognized as the same source during the fusion step, leading to an overestimation. As a result, compared to the basic density-clustering-based method, despite an advantage in the estimation accuracy, the probability distribution of estimated results is much better for the multimodal-fusion-based method. In addition, the estimation accuracy varies with the number of modes employed. Therefore, it can be seen that the presetting of the number of employed modes is quite important.

In contrast, the local-confidence-level-enhanced density clustering dramatically improves the accuracy of the source counting. In the enhanced method, the local density of the samples around the cluster centers with high local confidence is increased. Therefore, the clusters become more compact, which improves the source counting accuracy.

In order to evaluate the effect of the noise on the proposed method, an experiment with different SNRs was performed. The real SNR of the lake trial is unknown. It was assumed, empirically, to be 24 dB. The SNR was adjusted by adding simulated Gaussian white noise to the lake trial signal. The performance of the basic density-clustering-based method and the enhanced density-clustering-based method for 8, 12, 16, 20, and 24 dB (without simulated noise) is displayed in Figure 12 comparatively. As the SNR decreases, the performance of both methods degenerates gradually. However, the enhanced density-clustering-based method performs better than the basic density-clustering-based method and undergoes slower degradation.

To investigate the frequency bands of the targets in the experiment, each TF point was assigned to a target according to its dominant signal. If the DOA estimation of a certain TF point was 2° away from the nearest cluster center, it was assigned to this target. Otherwise, it is considered that there is no dominant signal in this time–frequency point. Figure 13 shows the assignment of the first 1.4 s. The frequencies of all four targets spread over the whole broadband of 500–8000 Hz randomly, and clearly cannot be divided into four different separated sub-bands. This result indicates that the WDO assumption holds in this experiment.

## 5. Conclusions

An underwater source counting method with local-confidence-level-enhanced density clustering is proposed. In this method, the STFT of the AVS output signal is calculated, and the DOAs of all time–frequency points consist of a data set. Then, the DOAs in the data set are classified with density clustering, where the cluster centers and the number of clusters are obtained as the DOA estimations and the target number, respectively. Finally, the local confidence level is applied to further enhance the performance of the density clustering algorithm. A lake trial is conducted to verify the local-confidence-level-enhanced source counting method which achieves a better source counting performance than the multimodal-fusion-based method and the enhanced density-clustering-based method.

## Figures and Tables

**Figure 1 sensors-23-08491-f001:**
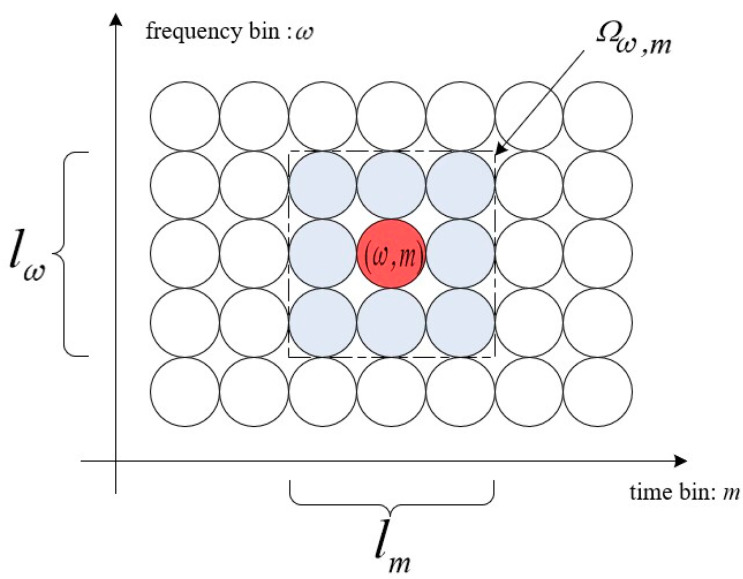
Time–frequency point (ω,m) and the surrounding rectangle area $\Omega_{\omega ,m}$.

**Figure 2 sensors-23-08491-f002:**
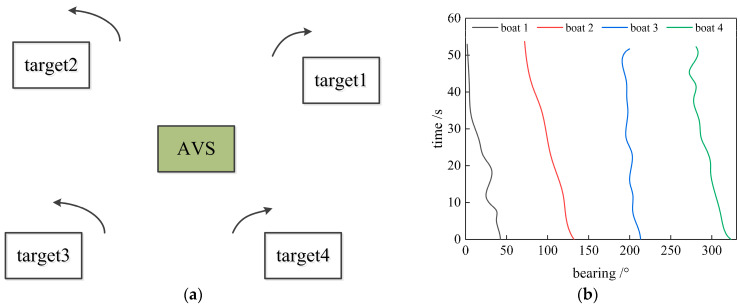
Experimental setup and traces of 4 boats: (**a**) Experimental setup; (**b**) Experiment azimuth waterfall map.

**Figure 3 sensors-23-08491-f003:**
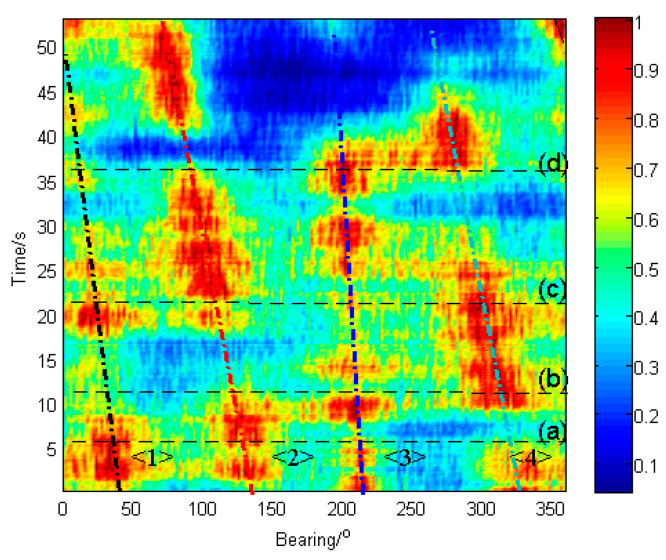
The temporal course of the cross-spectral DOA histogram of vector hydrophone from lake trial. ((**a**), (**b**), (**c**) and (**d**) denote the time of 6.7 s, 11.6 s, 21.3 s and 36.0 s).

**Figure 4 sensors-23-08491-f004:**
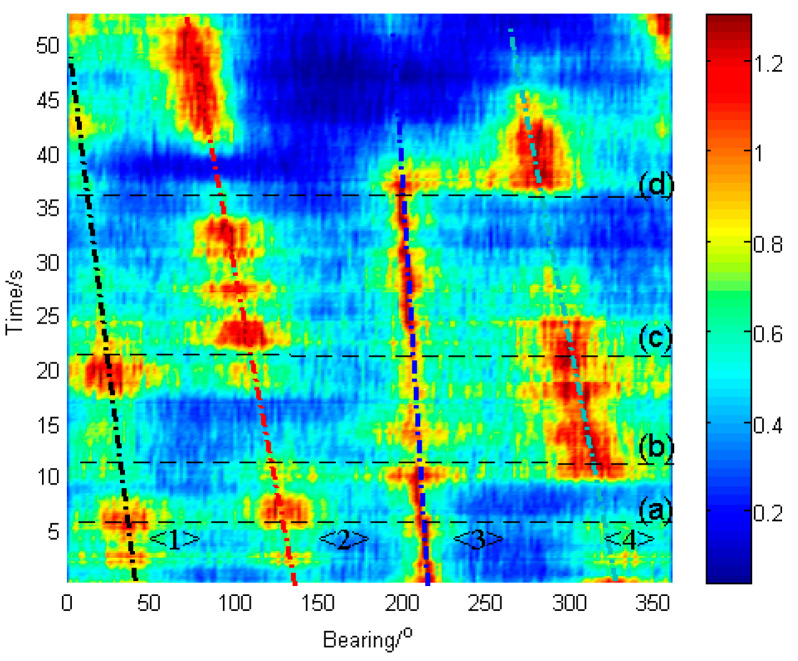
The temporal course of the local-confidence-level-enhanced cross-spectral DOA histogram. ((**a**), (**b**), (**c**) and (**d**) denote the time of 6.7 s, 11.6 s, 21.3 s and 36.0 s).

**Figure 5 sensors-23-08491-f005:**
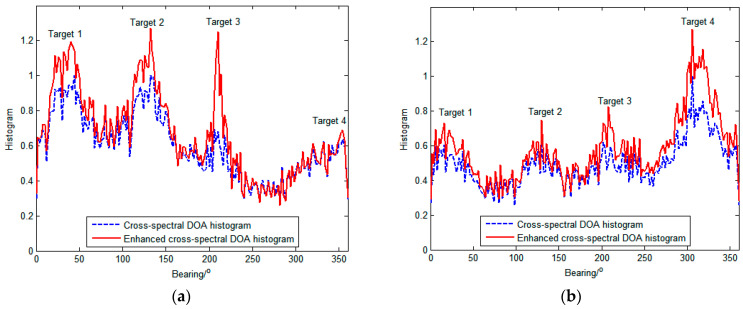
Comparison of cross-spectral DOA histograms and local-confidence-level-enhanced cross-spectral DOA histograms at different times. (**a**) DOA histograms at time (a); (**b**) DOA histograms at time (b); (**c**) DOA histograms at time (c); (**d**) DOA histograms at time (d).

**Figure 6 sensors-23-08491-f006:**
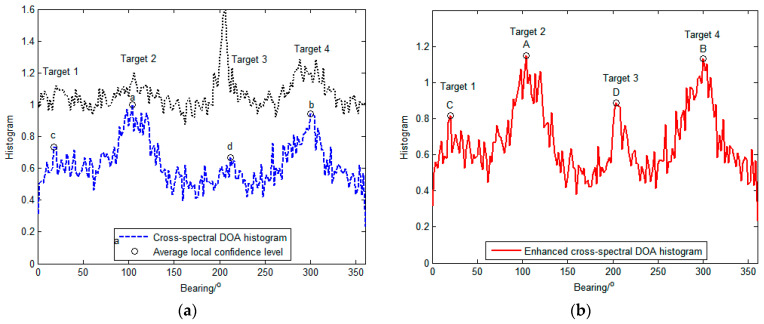
DOA histogram: (**a**) The cross-spectral DOA histogram and average local confidence level; (**b**) The enhanced cross-spectral DOA histogram. (The circles a–d and A–D denote the data samples considered as the clusters centers).

**Figure 7 sensors-23-08491-f007:**
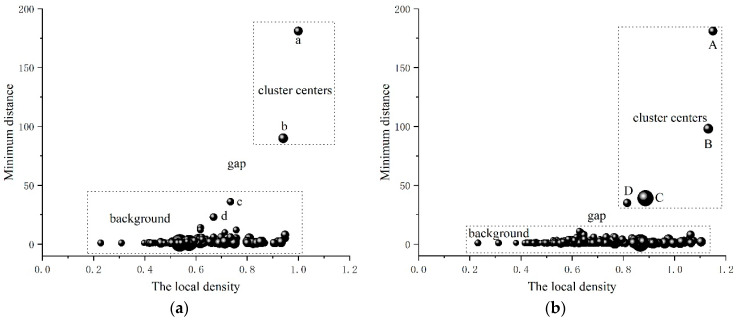
The decision graphs of the basic and enhanced density clustering algorithms for the data in Figure 6: (**a**) Basic density clustering; (**b**) Enhanced density clustering. (The points a–d and A–D denote the data samples considered as the clusters centers).

**Figure 8 sensors-23-08491-f008:**
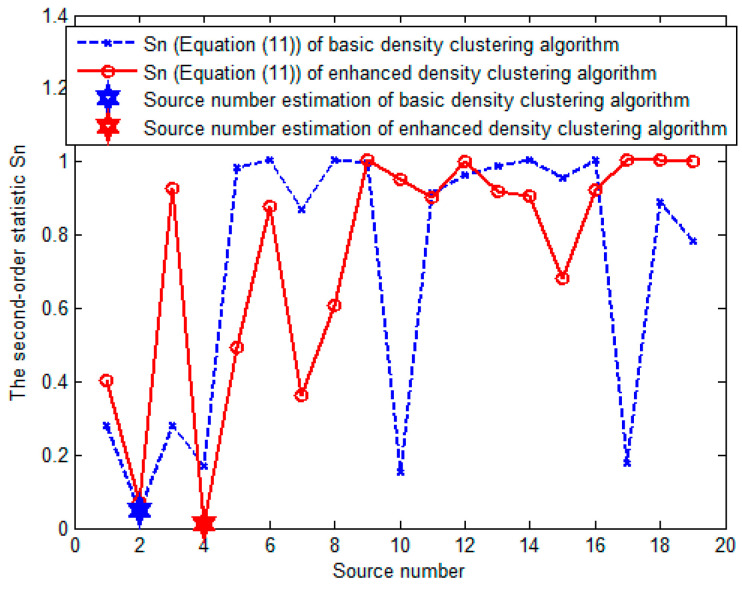
The second-order statistic $S_{n}$ of the basic and enhanced density clustering algorithms for the data in Figure 6.

**Figure 9 sensors-23-08491-f009:**
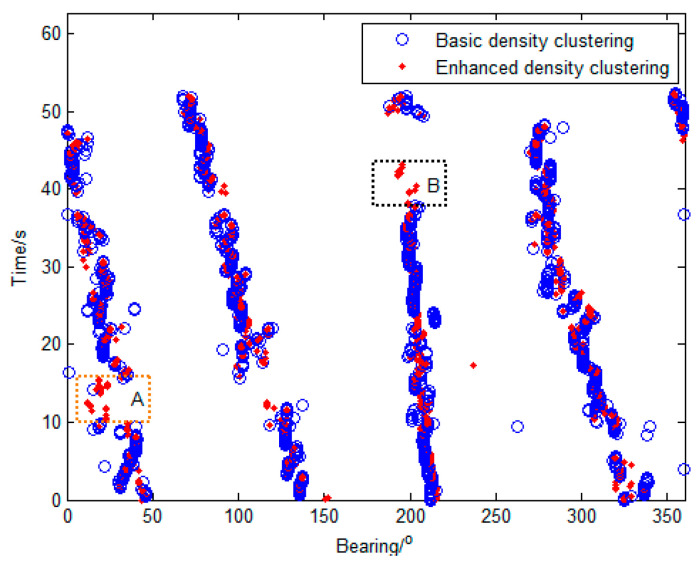
Courses of the multi-target DOA estimations with the basic density clustering, and the local-confidence-level-enhanced density clustering. (Comparing to the basic density clustering, the continuity of the courses of the enhanced density clustering is remarkably improved in Regions A and B).

**Figure 10 sensors-23-08491-f010:**
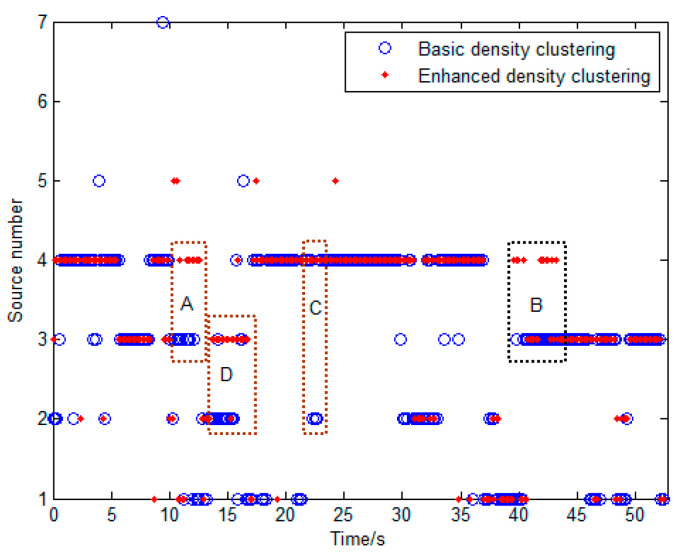
Courses of the source number estimations obtained by the basic density clustering and the local-confidence-level-enhanced density clustering. (The enhanced density clustering achieves the correct estimation results while the basic density clustering does not in Regions A, B, and C. The outcomes of the enhanced density clustering are closer to the true value in Regions D).

**Figure 11 sensors-23-08491-f011:**
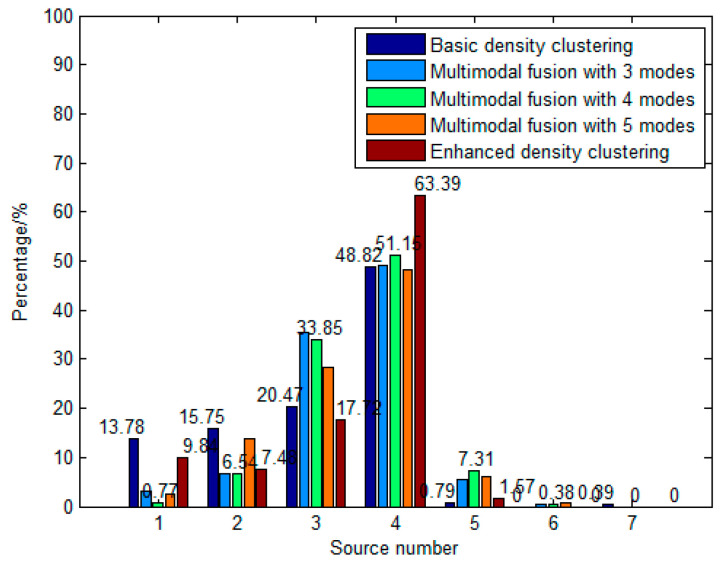
The estimated source number histogram of the basic density-clustering-based method, the multimodal-fusion-based method, and the enhanced density-clustering-based method for the experiment.

**Figure 12 sensors-23-08491-f012:**
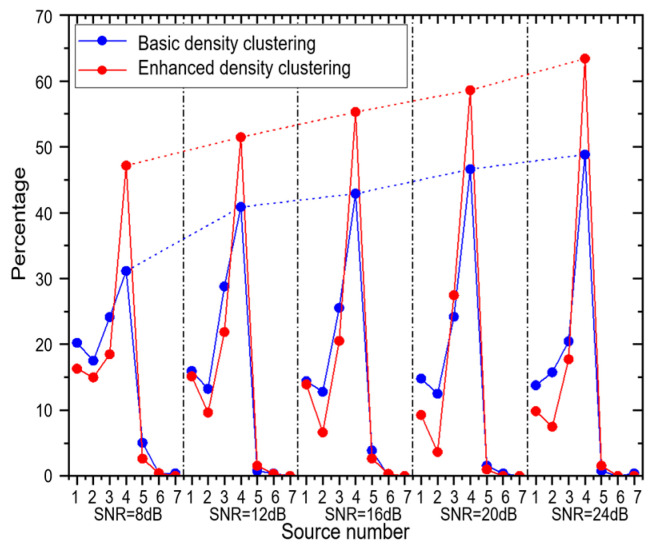
Source counting performance vs. SNR.

**Figure 13 sensors-23-08491-f013:**
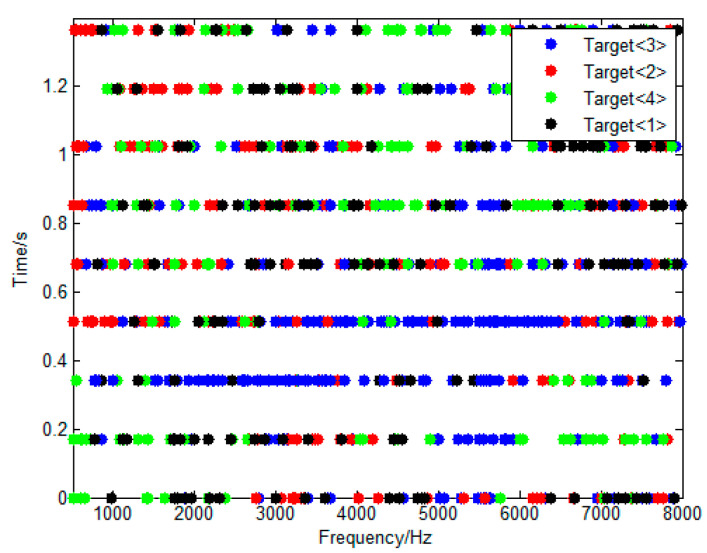
Frequency distribution of targets in time–frequency domain.

## Data Availability

Not applicable.

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
