# Peer review of "Underwater Source Counting with Local-Confidence-Level-Enhanced Density Clustering"

_sensors, 2023, doi:10.3390/s23208491_

Round 1

Reviewer 1 Report

1.   The authors propose a source counting method with local confidence level enhanced density clustering using a single acoustic vector sensor (AVS).  The short-time Fourier transforms (STFT) of the sound pressure and vibration velocity measured by the AVS are first calculated, and a data set is established with the direction of arrivals (DOAs) estimated from all of the time-frequency points.

2.      Please compare the contributions of the proposed technology to related technologies, in detail.

3.      Please elaborate the technique efficacy of the proposed method in detail.

4.      Please elaborate the concept of the equations 4, 5, 6, 7 and 14 in detail.

5.     In the figure 1, time-frequency point and the surrounding rectangle area, should be elaborated in detail.

6.    In the figure 3, cross-spectral DOA histogram of vector hydrophone from Lake trail, should be elaborated in detail.

7.   In the figure 4, the local confidence level enhanced cross-spectral DOA histogram, should be elaborated in detail.

Minor editing of English language required.

Reviewer 2 Report

Hope the authors can make the following supplementary modifications:

1. How does this approach relate to the signal-to-noise ratio?

2. If the signal strength of multiple targets is weak, is this method applicabl?

3.When multiple targets have the same signal frequency, a single vector hydrophone cannot distinguish the target. What are the frequencies of the four targets in the experiment?

4. How does the sound source count relate to the accuracy of each DOA in the data set?

Round 2

Reviewer 1 Report

No further comment.